# Gamma delta T cells recognize haptens and mount a hapten-specific response

Xun Zeng[1†], Christina Meyer[2†§], Jun Huang[1‡¶], Evan W Newell[1‡#], Brian A Kidd[1‡‖], Yu-Ling Wei[1], Yueh-hsiu Chien[1,2]*

[1]Department of Microbiology and Immunology, Stanford University, Stanford, United States; [2]Program in Immunology, Stanford University, Stanford, United States

*For correspondence: chien@
stanford.edu

[†]These authors contributed
equally to this work

[‡]These authors also contributed
equally to this work

Present address: [§]Department
of Immuno-Oncology, EMD
Serono Research and
Development Institute, Billerica,
United States; [¶]Institute for
Molecular Engineering,
University of Chicago, Chicago,
United States; [#]Singapore
Immunology Network,
Singapore, Singapore;
[‖]Department of Genetics and
Genomic Sciences, Icahn
Institute for Genomics and
Multiscale Biology, Ichan School
of Medicine at Mount Sinai, New
York, United States

Competing interests: The
authors declare that no
competing interests exist.

Reviewing editor: Xuetao Cao,
Zhejiang University School of
Medicine, China

**Abstract** The ability to recognize small organic molecules and chemical modifications of host molecules is an essential capability of the adaptive immune system, which until now was thought to be mediated mainly by B cell antigen receptors. Here we report that small molecules, such as cyanine 3 (Cy3), a synthetic fluorescent molecule, and 4-hydroxy-3-nitrophenylacetyl (NP), one of the most noted haptens, are γδ T cell antigens, recognized directly by specific γδ TCRs. Immunization with Cy3 conjugates induces a rapid Cy3-specific γδ T cell IL-17 response. These results expand the role of small molecules and chemical modifications in immunity and underscore the role of γδ T cells as unique adaptive immune cells that couple B cell-like antigen recognition capability with T cell effector function.

## Introduction

The adaptive immune system consists of B cells, αβ T cells and γδ T cells. While αβ T cells perform all well-defined functions attributed to T cells, γδ T cells and αβ T cells are present together in all but the most primitive vertebrates. This suggests that each cell type performs unique functions and that both are necessary for host immune competence. Indeed, although γδ T cells and αβ T cells have similar effector functions, γδ T cells and αβ T cells are distinct in their antigen recognition and activation requirements and in their antigen-specific repertoire and effector function development. These differences underlie γδ T cells' unique contribution to host immune defense (*Chien et al., 2014*).

Diversity in antigen receptor specificity is the hallmark of the adaptive immune system. Serological analysis of small chemical compound immune recognition was one of the earliest experimental demonstrations that B cells can mount responses to diverse antigens with specificity (*Landsteiner and van der Scheer, 1931*; *Landsteiner and Chase, 1937*). Haptens were characterized as small organic molecules which, when conjugated to a protein, induce a strong hapten-specific B cell response. Since then, antibody responses to haptens have been used extensively to investigate antibody affinity maturation, germinal center formation, and the development of memory B cell responses (*Jack et al., 1977*; *Jacob et al., 1991*; *McHeyzer-Williams and McHeyzer-Williams, 2005*). Antibodies specific for pathogen-produced small compounds and chemical modifications of host molecules have also served as a means of pathogen surveillance (*Daneshvar et al., 1989*; *Temmerman et al., 2004*) and to monitor injury or altered physiological states (*Vossenaar et al., 2004*; *Kim et al., 2006*; *Yang and Sauve, 2006*). Thus, small molecule recognition is an important capability of the adaptive immune system.

Although hapten-specific αβ T cells have been reported and studied in the context of suppressor T cell function, as exemplified by the work of Dorf et al. (*Sherr and Dorf, 1981*), interaction between the T cell receptor (TCR) and the hapten ligand has not been demonstrated. Moreover, it is well established that the antigen-specific repertoires of peripheral αβ T cells are largely limited to peptides that are processed from protein antigens in complex with the host major histocompatability complex (MHC) molecules (*Huseby et al., 2005*; *Van Laethem et al., 2007*). This restriction on antigen specificity

**eLife digest** Our immune system responds to invading microbes—such as viruses and bacteria—and tries to eliminate the threat via two distinct but connected systems: the innate and the adaptive immune systems. Cells of the innate immune system patrol our organs and tissues in an effort to identify and eliminate threats with a quick but general response, which is similar for many different pathogens. This first line of defense also escalates the immune response by activating the adaptive immune system.

Unlike the innate immune response, the adaptive immune response targets unique molecules of different sizes, shapes and chemical compositions—ranging from small organic molecules to large pathogens. The adaptive immune system consists of three types of immune cells: B cells, alpha beta (αβ) T cells and gamma delta (γδ) T cells. These cells have proteins on their surfaces that function as receptors; when the receptors recognize and bind to a foreign molecule (called antigen), the cell becomes activated. This then triggers a cascade of events that help to clear the infection and help immune cells to rapidly respond to any future infection by the same pathogen. αβ T cells and γδ T cells respond to different triggers, but perform similar tasks—while B cells perform tasks that are different from those of T cells. An effective immune response often involves both B cells and T cells.

One important way that the adaptive immune system can identify an invading microbe or monitor for damaged or abnormal cells is by recognizing chemicals produced by pathogen and chemical modifications of host molecules. And while B cells are able to do this, αβ T cells are not.

Zeng et al. now show that γδ T cells can also recognize and mount response against this type of antigen. γδ T cells were shown to detect both a small synthetic fluorescent dye, and a chemical modification that has been extensively studied for B cell responses over the last 80 years. Following on from these findings, the next challenge is to identify γδ T cells that recognize molecules or chemical compounds produced during infection or disease, and to define these cells' role in immunity.

is a consequence of the thymic development process (*Van Laethem et al., 2012*). Thus, adaptive immune recognition of small molecules seems to be mainly mediated by B cells rather than T cells.

While γδ T cells, like αβ T cells, require thymic maturation before entering the periphery (*Ohno et al., 1993*), this process does little to constrain the γδ T cell antigen-specific repertoire (*Jensen et al., 2008*). In addition, although fetal-derived γδ T cells in murine skin and the reproductive tract express non-variant TCRs, adult human and murine γδ T cells in other lymphoid compartments (blood, lymph node, spleen, and intestine) express diverse TCRs (*Chien et al., 2014*). Analysis of γδ TCR CDR3 sequence diversity and length distribution suggest that these T cells have extensive antigen recognition capability and that as a group, γδ TCRs are more similar to immunoglobulins (Igs) than to αβ TCRs (*Rock et al., 1994*). Since the requirements of γδ T cell antigen recognition are similar to those of B cells, we investigated whether γδ T cells, like B cells, can recognize haptens.

Here, we report that Cyanine 3 (Cy3), a synthetic fluorescent molecule, is a γδ T cell antigen, recognized directly by specific γδ TCRs. Immunization with Cy3 induces γδ T cells to mount a Cy3-specific IL-17 response. IL-17 is a T cell cytokine, which is essential in the initiation of the inflammatory response. We also identified γδ TCRs specific for 4-hydroxy-3-nitrophenyl acetyl (NP), one of the most commonly studied haptens in investigation of antibody response. These results enlarge the scope of the γδ T cell antigen-specific repertoire and suggest a way for this category of antigens to induce a T cell response.

## Results

### Cyanine 3 (Cy3) is a γδ T cell antigen

To test whether γδ T cells can recognize small molecules, we chose Cyanine 3 (Cy3) for analysis. Cy3 is a synthetic dye with two modified indole groups joined by a polymethine chain (*Figure 1A*). It is highly fluorescent and can be used for FACS analysis directly.

Cy3 tetramer ($Cy3_4SAv$) (a recombinant mutant of streptavidin (*Ramachandiran et al., 2007*) labeled with four Cy3 molecules at the C-terminal cysteine in each of the four identical subunits)

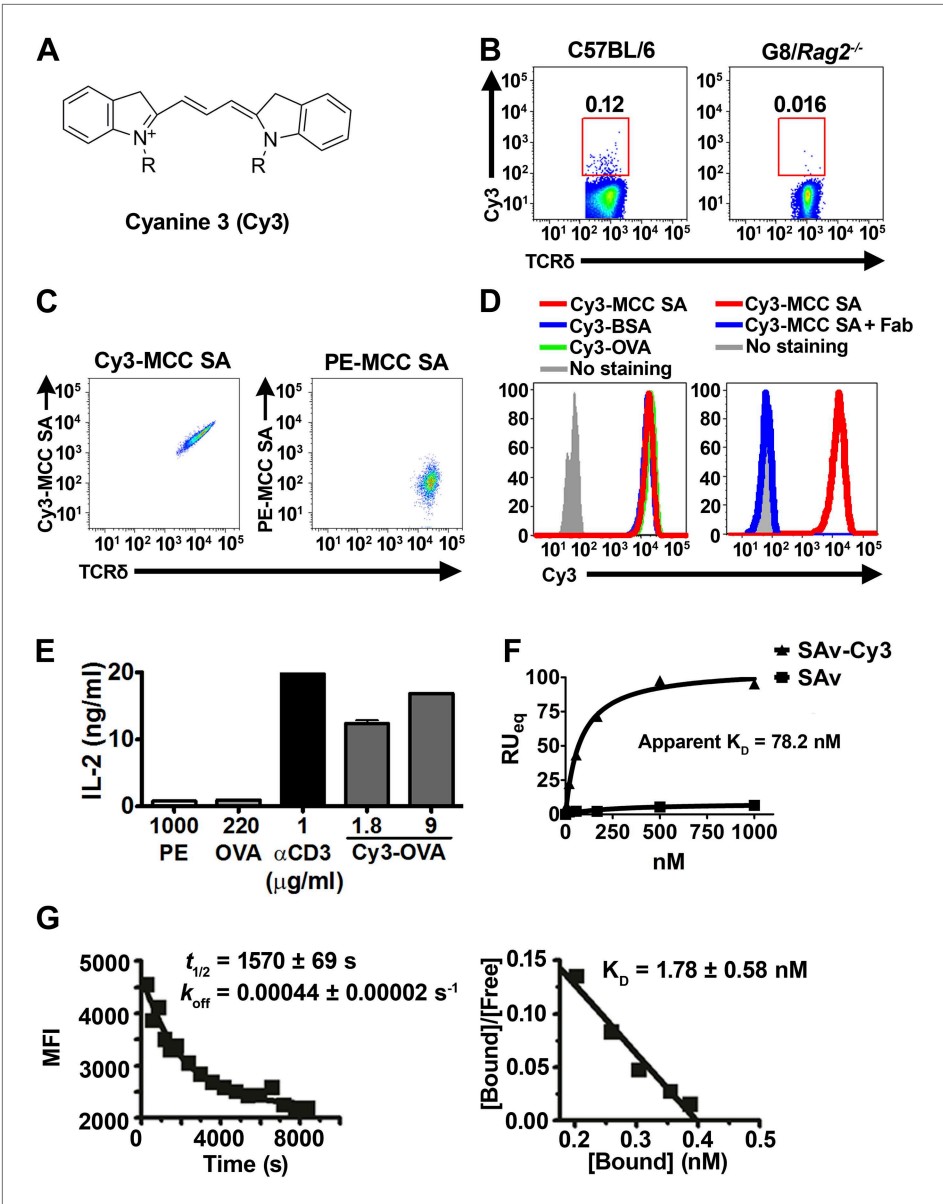

**Figure 1**. Cy3 is a γδ T cell antigen. (**A**) Chemical structure of Cyanine 3 (Cy3). FACS analysis of (**B**) Cy3 tetramer (Cy3$_4$-SAv) staining of splenic γδ T cells in the presence of 10-fold molar excess of moth cytochrome c peptide coupled SAv (MCC$_4$-SAv); (**C**) NX6/58α-β- cells stained with Cy3-MCC-SAv or PE-MCC-SAv; (**D**) NX6/58α-β- cells stained with Cy3-MCC-SAv in the absence (left), or presence of anti-Cy3 Fab (right). (**E**) IL-2 production by NX6/58α-β- cells activated by the indicated amount of plate-bound Cy3-OVA, OVA, PE, anti-CD3 for 16 hr. (**F**) The saturating binding curves of Cy3$_4$-SAv and un-conjugated SAv to a soluble form of NX6 as determined by surface plasmon resonance. No detectable binding was observed for 1 mM applications of PE or BSA (not shown). (**G**) Kinetics of Cy3$_4$SAv binding to NX6/58α-β- cells. $t_{1/2}$ was determined using real time flow cytometry in the presence of anti-Cy3 antibody Fab fragments (left). K$_D$ was determined from Scatchard analysis (right). All results are representative of at least three independent experiments.

The following figure supplements are available for figure 1:

**Figure supplement 1**. NX6/58α-β- cells stained with different fluorescently labeled ovalbumin preparations.

**Figure supplement 2**. Correlation between the mean fluorescence intensities of PE-SAv and Cy3$_4$SAv on red blood cells.

stained ~0.05–0.2% of normal splenic γδ T cells, but not G8/*Rag2*$^{-/-}$ γδ TCR transgenic cells (specific for the nonclassical MHC class I T10 and T22) (*Bluestone et al., 1988*; *Schild et al., 1994*; *Figure 1B*). We then identified Cy3-specific γδ TCRs on a single cell level by sorting these cells and sequencing their TCR genes. 58α-β- cells expressing Cy3-specific γδ TCRs bound Cy3-ovalbumin (Cy3-OVA), Cy3-bovine serum albumin (Cy3-BSA), Cy3-MCC-streptavidin (moth cytochrome C (MCC)-derived peptide, Cy3-labeled at the N-terminus, biotinylated at the C-terminus, and tetramerized with streptavidin), but not FITC or APC labeled OVA, nor PE-MCC peptide/streptavidin (*Figure 1C*, *Figure 1—figure supplement 1*; *Table 1*). Moreover, Cy3-MCC-streptavidin staining of a Cy3-specific γδ TCR NX6/58α-β- was inhibited by the inclusion of Fab fragments of an anti-Cy3 antibody (clone A-6; Santa Cruz Biotechnology) (*Figure 1D*). In addition, NX6/58α-β- cells were activated by plate-bound Cy3-OVA, but not unmodified OVA (*Figure 1E*). Binding of the soluble form of a Cy3-specific γδ TCR (NX6) to Cy3$_4$SAv can be demonstrated by surface plasmon resonance (Biacore) with an apparent $K_D$ of 78.2 nM (*Figure 1F*). We also examined the affinity of Cy3$_4$SAv binding to NX6 expressed on 58α-β- cells. Scatchard analysis showed an apparent nanomolar $K_D$ (1.8 nM) with a half-life of ~26 min (*Figure 1G*). Taken together, these results indicate that Cy3 is an antigen of γδ T cells, recognized directly by specific γδ TCRs.

## γδ T cells mount a hapten-specific response

To determine whether γδ T cells can mount a hapten-specific response, we immunized mice subcutaneously with Cy3–chicken gamma globulin (Cy3-CGG) in aluminum hydroxide (alum) and analyzed Cy3-specific γδ T cells in the draining lymph nodes with a Cy3-OVA staining reagent. For comparison, we also analyzed Cy3-specific γδ T cells in mice immunized with CGG/alum. Alum was used because it is a non-antigenic adjuvant (*Eisenbarth et al., 2008*), and we chose subcutaneous immunization because it focuses the immune response to the draining lymph nodes.

We found that prior to immunization, ~80% of Cy3-specific γδ T cells in the lymph nodes were CD44$^{lo}$, a phenotype typical of naïve T cells. Within 24 hr after immunization, Cy3-specific γδ T cells up-regulated CD44 in Cy3-CGG-immunized mice, but not in CGG-immunized mice (*Figure 2A*). BioMark analysis showed that Cy3-specific γδ T cells express the mRNA coding for RORγt, IL-17A and IL-17F 60 hr after immunization (*Figure 2B*). Consistent with this observation, analysis of Cy3-specific γδ T cell responses in IL-17F reporter mice (*Il-17f*$^{Thy1.1/Thy1.1}$) (*Lee et al., 2009*) and staining showed that 60 hr after Cy3-CGG immunization, activated Cy3-specific γδ T cells expressed the Thy1.1 reporter or IL-17 protein (*Figure 2C*). In addition, we found that activated Cy3-specific γδ T cells expressed the receptors for IL-1 and IL-23 (*Figure 2B*), a characteristic similar to our analysis of activated PE-specific γδ T cells in an immune response (*Zeng et al., 2012*). The expression of inflammatory cytokine receptors allows antigen-activated γδ T cells to integrate signals from antigen receptors and cytokine receptors to mount an enhanced and sustained response (*Zeng et al., 2012*).

Taken together, the observations that Cy3-specific γδ T cells can be activated and produce IL-17 upon Cy3-CGG, but not CGG, immunization indicates that γδ T cells, like B cells, are capable of mounting specific responses to small molecules.

## 4-hydroxy-3-nitrophenyl acetyl (NP) is a γδ T cell antigen

To test the generality of the observation that γδ T cells can recognize small molecules, we chose 4-hydroxy-3-nitrophenyl acetyl (NP) for analysis. NP is one of the most commonly studied hapten

**Table 1.** TCR sequences of Cy3 and NP-specific γδ TCRs

| | | | Vδ | N | D1 | N | D2 | N | Jδ | | Vγ | N | Jγ |
|---|---|---|---|---|---|---|---|---|---|---|---|---|---|
| Cy3 | NX6 | Vδ8 | C A A S | | | | | A | T D K | | Vγ1 | C A V W | S R | S G T S W V K |
| | C5 | Vδ6A | C A L W E L | G | | | G G I R | A S | D K | | Vγ1 | C A V W | T R | G T S W V K |
| NP | 1G9 | Vδ4 | C A L M E R | R | | G Y | R R D T | R A | D K | | Vγ4 | C S Y G S | Y | S S G F H K |
| | 1E3 | Vδ6B | C A L S E L | G G | | | G G | S A | T D K | | Vγ1 | C A V W | K K T | G T S W V K |
| | 1B2 | Vδ4 | C A L M E R | V | G | L Y | R R D T | S L A | T D K | | Vγ1 | C A V | F | S G T S W V K |

Each pair of γ and δ chain sequences were identified from a single Cy3 or NP-specific γδ T cell derived from mouse splenocytes and verified by their ability to confer NP- or Cy3-specific binding to 58α-β- cells expressing the TCR.

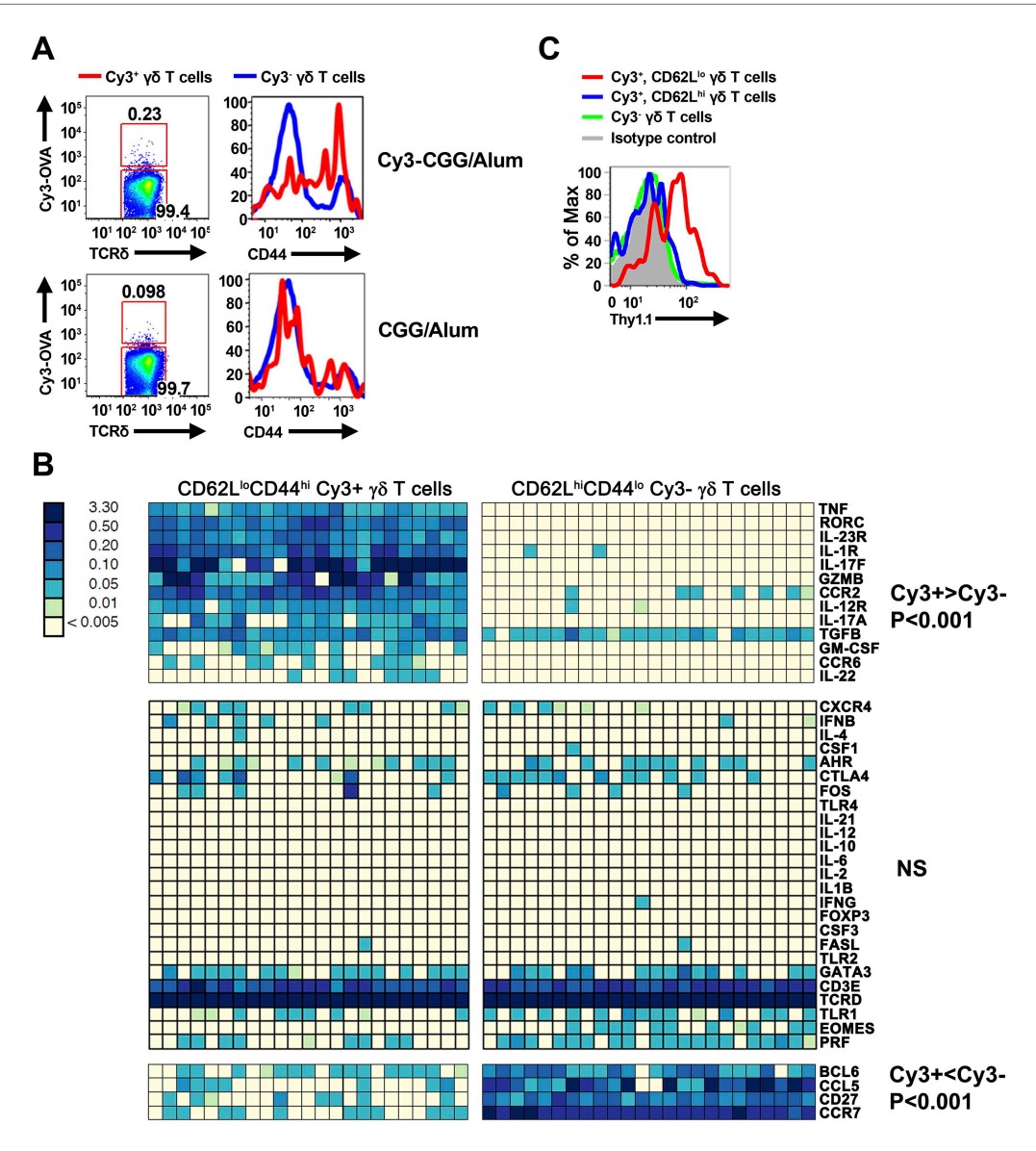

**Figure 2**. Cy3-specific γδ T cell response after immunization. (**A**) CD44 expression on Cy3-OVA⁺ (red) and Cy3-OVA⁻ γδ T cells in the draining lymph nodes of mice immunized with Cy3-CGG-alum or CGG-alum 24 hr prior. (**B**) BioMark analysis of CD62L$^{lo}$CD44$^{hi}$ Cy3⁺ and CD62L$^{hi}$CD44$^{lo}$ Cy3⁻ γδ T cells isolated from the draining lymph nodes of C57BL/6 mice immunized with Cy3-CGG 60 hr prior (5 cells/sample). The heatmap, where rows are individual genes and columns are individual samples, indicates the expression or non-expression of a gene/sample pair (relative to the *β2m* expression). Upper panel shows genes expressing higher (p < 0.001) in Cy3⁺ cells than that in Cy3⁻ cells. Middle panel shows non-varying genes. Bottom panel shows genes expressing lower (p < 0.001) in Cy3⁺ cells than that in Cy3⁻ cells. (**C**) Thy1.1 expression on γδ T cells from *IL-17f*$^{Thy1.1/Thy1.1}$ mice immunized with Cy3-CGG-alum 60 hr prior, representative of three independent experiments.

molecules in investigations of antibody responses (*Jack et al., 1977*; *Jacob et al., 1991*; *McHeyzer-Williams and McHeyzer-Williams, 2005*), and NP is structurally unrelated to Cy3 (*Figure 3A*).

NP conjugated to a fluorescent protein, phycoerythrin (PE), is routinely used to identify NP-specific B cells in FACS analysis. We found that NP-PE stained ~0.14% of splenic γδ T cells of normal mice (left panel), but not G8/*Rag2*⁻/⁻ γδ TCR transgenic cells (middle panel). Consistent with the observation that PE is a γδ T cell antigen (*Zeng et al., 2012*), we found ~0.03% of splenic γδ T cells stained with PE under the same staining conditions (right panel). After accounting for background

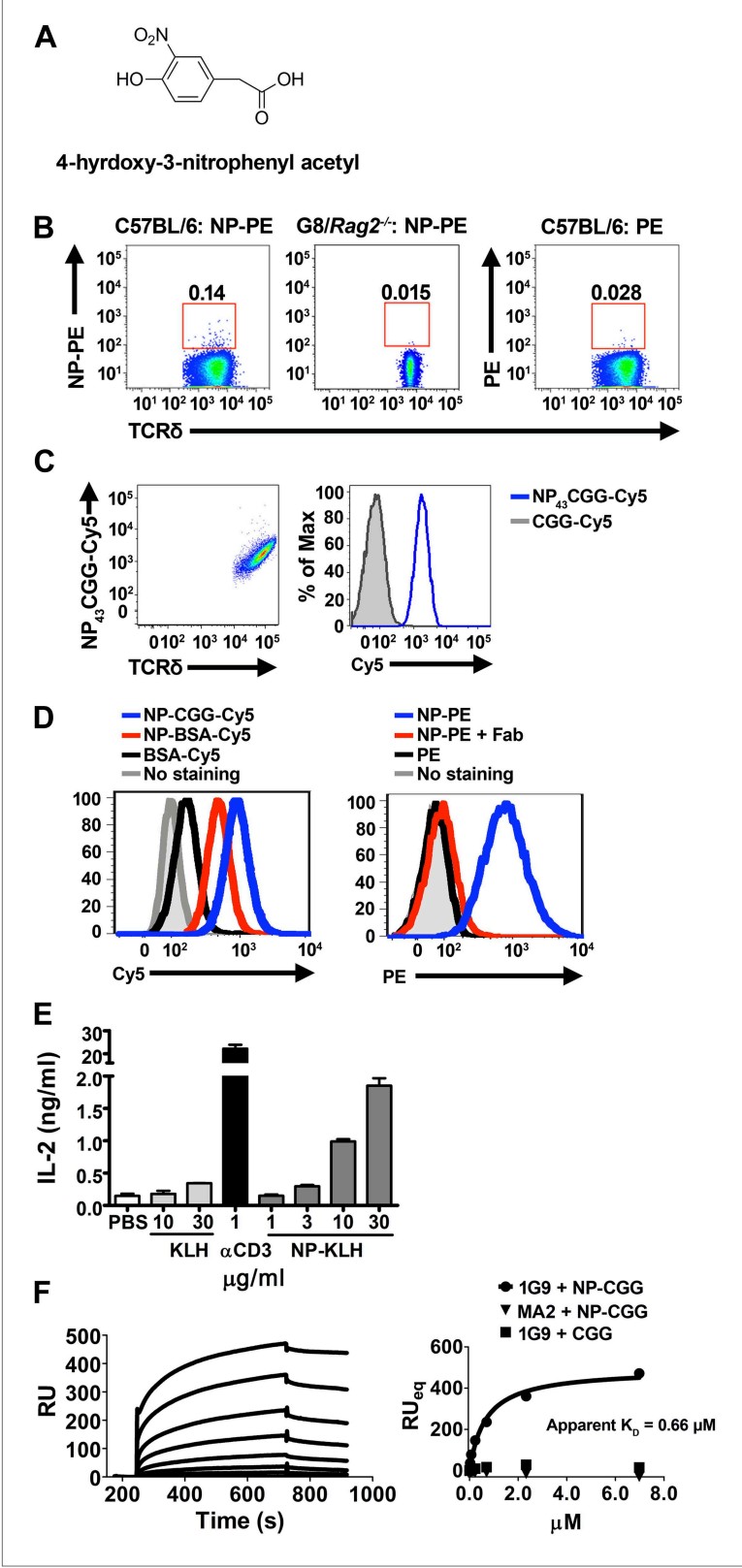

**Figure 3**. NP is a γδ T cell antigen. (**A**) Chemical structure of 4-hydroxy-3-nitrophenyl acetyl (NP). Flow cytometry analysis of (**B**) NP$_{67}$-PE staining of γδ T cells from C57BL/6 or G8/*Rag2*$^{−/−}$ mouse splenocytes and PE staining of γδ T cells from B6 splenocytes; (**C**) staining of 58α-β- cells expressing an NP-specific γδ TCR, 1G9, with NP$_{43}$-CGG-Cy5 or
*Figure 3. Continued on next page*

*Figure 3. Continued*

CGG-Cy5, showing staining in relation to γδ TCR expression (left) or as a histogram (right); (**D**) staining of 58α-β- cells expressing an NP-specific γδ TCR, 1E3, with NP$_{43}$-CGG-Cy5, NP$_{26}$-BSA-Cy5, or BSA-Cy5 (left) and NP$_{67}$-PE alone, NP$_{67}$-PE with a 20-fold molar excess of anti-NP Fab, or PE (right). (**E**) IL-2 production by 1E3/58α-β- cells activated by the indicated amount of plate-bound NP$_{25}$-KLH, KLH (light gray bars), or 0.1 µg/ml anti-CD3. (**F**) Sensorgram and steady state analysis of NP$_{43}$-CGG (0–7 µM) binding to soluble 1G9 TCR measured by surface plasmon resonance. Apparent $K_D$ was determined by steady state analysis of SPR measurements (circles). Equal concentrations of un-modified CGG were tested (squares), as well as NP$_{43}$-CGG with a PE-specific γδ TCR, MA2 (triangles).

staining and for PE staining, we estimated that ~0.1% of total γδ T cells could be NP-specific (*Figure 3B*).

We further identified NP-specific γδ TCRs on a single cell level. Expressing NP-specific γδ TCRs in 58α-β- cells enables these cells to be stained with NP-CGG-Cy5, but not CGG-Cy5 (*Figure 3C*; *Table 1*). Further investigation showed 58α-β- cells expressing NP-specific γδ TCRs could also be stained with NP-BSA-Cy5 and NP-PE, but not with BSA-Cy5 or PE (*Figure 3D*, left panel). In addition, NP-PE staining was inhibited by the inclusion of Fab fragments of an anti-NP antibody (clone H33Lγ; G. Kelsoe) (*Figure 3D*, right panel). Furthermore, 58α-β- cells expressing NP-specific γδ TCRs produced IL-2 in response to plate-bound NP-keyhole limpet hemocyanin (NP-KLH), but not plate-bound KLH in a dose-dependent manner (*Figure 3E*). The observations that only molecules containing the NP conjugation stain NP-specific γδ TCR-expressing cells, that NP-conjugate staining is blocked by an anti-NP Fab, and that an immobilized NP-conjugate can activate NP-specific γδ T cells indicate that NP is recognized directly by specific γδ TCRs. Indeed, direct binding between soluble NP-specific γδ TCRs (1G9) and NP-conjugates was also demonstrated using surface plasmon resonance (*Figure 3F*). The measured apparent $K_D$ for the interaction between NP$_{43}$-CGG and the 1G9 TCR was 0.66 µM. NP$_{43}$-CGG exhibited no binding to the PE-specific γδ TCR, MA2 (*Zeng et al., 2012*), and CGG did not bind 1G9 (*Figure 3F*). Taken together, these results show that NP is a γδ T cell antigen and is recognized directly by specific γδ TCRs.

## Discussion

At the turn of the last century, Landsteiner pioneered the use of small synthetic molecules, known as haptens, to induce an antibody response. When coupled with carrier proteins, haptens induce a strong (hapten) specific, (αβ) T cell-dependent B cell response. Since the hapten modification provides a defined epitope for analysis of the antibody response, haptenated proteins have been used extensively to characterize the development of B cell responses, and NP is one of the most commonly studied haptens. Although Cy3 has not been used in this context previously, high affinity, isotype-switched Cy3-specific antibodies are widely available commercially, indicating that Cy3 is also highly immunogenic, similar to other well-studied hapten molecules. Our demonstration that γδ T cells can directly recognize and respond to these molecules represents a significant expansion in the scope of the γδ T cell antigen-specific repertoire.

This is the first demonstration that γδ TCRs can interact directly with small molecules. In this context, prior work has shown that a collection of small pyrophosphate-containing organic molecules can stimulate human Vγ9Vδ2-expressing γδ T cells (also referred to as Vγ2Vδ2 by the Seidman et al. nomenclature) in vitro in a TCR-dependent manner (*Chien et al., 2014*). These molecules, collectively known as phosphoantigens (pAgs) include isopentenyl pyrophosphate (IPP), an intermediate of the human mevalonate pathway, and (E)-4-hydroxy-3-methyl-but-2-enyl-pyrophosphate (HMBPP), a microbial isoprenoid intermediate. However, recent reports indicate that pAgs do not interact directly with the γδ TCR. Instead, Vγ9Vδ2 T cell activation by pAgs is through the recognition of an allosteric change in the extracellular domain of a cell surface molecule, butyrophilin 3A1, which is induced in response to intracellular accumulation of pAg (*Wang et al., 2013*; *Sandstrom et al., 2014*).

Our past studies indicate that γδ T cells need not encounter cognate antigen in the thymus to signal through the TCR, mature, and exit to the periphery. Peripheral γδ T cells derived from γδ thymocytes that have not previously encountered thymic ligands produce IL-17 upon TCR triggering (*Jensen et al., 2008*). Indeed, we have identified multiple foreign molecules which are γδ T cell antigens:

phycoerytherin (PE), a member of the phycobiliprotein family, which is located on the tip of photosynthetic antenna of red algae and cyanobacteria (*Zeng et al., 2012*), and here, the haptens Cy3 and NP as γδ T cell antigens. Moreover, both Cy3 and PE-specific γδ T cells differentiate toward an IL-17-producing phenotype with similar activation kinetics upon antigen encounter (*Zeng et al., 2012*) (and this manuscript): within 24 hr after immunization, PE- or Cy3 specific γδ T cells in the draining lymph node showed activated phenotypes, such as becoming CD44$^{hi}$ and CD62L$^{lo}$. Activated antigen-specific γδ T cells express RORγt 48 hr after immunization and, after another 12 hr, IL-17A and IL-17F. Significantly, the expression of inflammatory cytokine receptors such as IL-1R and IL-23R are induced on antigen activated γδ T cells. The cytokine-receptor signaling provides a 'second signal' in addition to TCR engagement to perpetuate the response in inflammation (*Zeng et al., 2012*).

The inflammatory response is an essential mechanism in the host response to infection and injury. In vertebrates, it requires IL-17, a cytokine primarily made by T cells. IL-17 induces the maturation and release of neutrophils from the bone marrow (*Stark et al., 2005*). Neutrophil infiltration focuses the immune response at the site of infection or injury, where antigen-specific αβ T cells subsequently proliferate and gain effector functions after stimulation by professional antigen-presenting cells and a particular cytokine environment. In acute infection, the host must make IL-17 rapidly without prior antigen exposure. The results presented here, together with our previous studies, suggest that γδ T cells are uniquely suited for the initial IL-17 response and provide a way for haptens to elicit this important T cell response. In this context, it has been noted that haptenation enhances the immunogenicity of the carrier protein, but this effect is independent of innate immune recognition and signaling (*Palm and Medzhitov, 2009*).

While serological responses to haptens were first demonstrated to illustrate the capability of the immune system to recognize diverse antigens, it appears that adaptive immune recognition of hapten-like pathogen-derived organic compounds and chemical modifications of host molecules can serve as a means of pathogen surveillance and monitoring of injury or altered physiological states. The synthetic hapten molecule NP is structurally similar to nitrated tyrosine (3-NTyr). 3-NTyr-containing proteins are formed in the presence of peroxynitrite, one of the side products of reactive oxygen and nitrogen species produced during the early stages of inflammation (*Beckman et al., 1992*; *Ischiropoulos et al., 1992a*; *Ischiropoulos et al., 1992b*; *Beckmann et al., 1994*). Tyrosine nitration has been demonstrated in a variety of infectious and inflammatory contexts, such as *Trypanosoma cruzi* infection (*Naviliat et al., 2005*; *Dhiman et al., 2008*) and atherosclerosis (*Beckmann et al., 1994*). There have been reports indicating that these pathological processes are driven in part by IL-17 and γδ T cells (*Stemme et al., 1991*; *Kleindienst et al., 1993*; *Lima and Titus, 1996*; *Hashmi and Zeng, 2006*; *Sardinha et al., 2006*; *Cheng et al., 2008*; *van Es et al., 2009*). Furthermore, the synthetic hapten molecule Cy3 contains two modified indole groups joined by polymethine bonds. The indole molecule is a noted bacterial product and signaling molecule, which accumulates at the site of bacterial infection and affects antibiotic resistance and other virulence factors (*Martino et al., 2003*; *Lee et al., 2007*; *Hirakawa et al., 2009*; *Lee et al., 2010*; *Kim et al., 2011*). An indole group also forms the side chain of tryptophan. Altered tryptophan metabolism along the kynurenine pathway and an unrestrained γδ T cell IL-17 response were identified as the causes of lethal pulmonary aspergillosis in a mouse model of chronic granulomatous disease (*Romani et al., 2008*). Whether hapten-specific γδ T cells also recognize structurally similar natural products, such as 3-NTyr and indole groups, is unclear. Regardless, our observations that small molecules and chemical modifications on proteins are γδ T cell targets suggest a new category of antigen specificity in addition to cell surface molecules such as the non-classical MHC class I molecules T10 and T22, MHC class I-related chain A/B (MICA/B), and endothelial protein C receptor (EPCR) (*Willcox et al., 2012*) that can activate γδ T cells in infection and inflammation.

The role of γδ T cells in hapten-driven pathological situations is currently unclear, and with these new findings worthy of future study. Allergic contact dermatitis (ACD) represents a specific example of a delayed-type-hypersensitivity response with a hapten-driven mechanism. γδ T cells have been implicated in mouse models of ACD. Some studies suggest that γδ T cells assist αβ T cells in adoptive transfer of contact sensitivity (*Ptak and Askenase, 1992*), while others suggest that γδ T cells regulate effector αβ T cell responses (*Guan et al., 2002*). Given our findings that γδ T cells can recognize haptens and mount a hapten-specific immune response, studies of the role of hapten-specific γδ T cells in processes like ACD could yield interesting results.

Although diversity in antigen receptor specificities is the hallmark of the adaptive immune system, effective adaptive immune responses are focused in antigen specificity. This is best illustrated in αβ T

cell-dependent antibody responses, wherein only αβ T cells that can recognize proteins that are internalized and presented by B cells and displayed as peptide/MHC complexes on cell surface can provide B cell help. Thus, only haptens coupled to proteins, which can be processed and presented for αβ T cell recognition, can induce a hapten-specific antibody response. While αβ T cells are responsible for the development of high affinity, isotype-switched antibodies, we found that γδ T cells recognize and respond to noted B cell antigens such as PE, NP and Cy3. In addition, in a case of human autoimmune myositis, where clonally expanded γδ T cells destroy muscle fiber, the targets of γδ T cells were also the targets of autoantibodies known as anti-Jo-1 (*Bruder et al., 2012*). These observations indicate that an overlap between the γδ T cell and B cell antigen-specific repertoires. If the frequencies of other antigen-specific γδ T cells were also in a similar range as that of PE, Cy3 and NP, then the numbers of distinct γδ T cell antigens would be ~$10^3$–$10^4$. The size of the B cell antigen-specific repertoire was estimated as roughly $10^5$, based largely on antigen-specific B cell frequencies of 0.004–0.007% for nitrophenyl (NP), dinitrophenyl (DNP), and trinitrophenyl (TNP). These values were obtained using antigen-specific spleen foci formation, (*Press and Klinman, 1974*; *Stashenko and Klinman, 1980*) and are likely to be underestimates, as this assay requires extensive proliferation of individual clones. In fact, FACS analysis showed that in naïve mice, 0.1% of the B cells are PE-specific and 0.02% allophycocyanin (APC)-specific (*Pape et al., 2011*). Using these values, the size of the antigen-specific B cell repertoire would be ~1000–5000, in the same range as that estimated for γδ T cells. Regardless of the extent of overlap between B cell and γδ T cell antigen-specific repertoires, our results here support previous observations (*Bruder et al., 2012*; *Zeng et al., 2012*) that γδ T cells and β cells can recognise the same antigen. In particular, NP- and PE have been used extensively as model antigens to elucidate principles of antibody affinity maturation, germinal center formation and the development of memory B cell responses. These studies should provide a context to study the roles of γδ T cells in the development of an integrated adaptive immune response.

## Materials and methods

### Reagents, mice, and immunization

Cy3 labeling of biotinylated moth cytochrome c (MCC) peptide (residues 88–103), ovalbumin (OVA) (Sigma, St. Louis, MO), BSA (Sigma), CGG (EMD Millipore, Billerica, MA), and streptavidin (SAv) was carried out with Cy3 maleimide and amine-reactive labeling kits (GE Healthcare, Little Chalfont, UK). NP (4-hydroxy 3-nitrophenylacetyl)-phycoerythrin (PE) was prepared using NP-O succinymidyl ester (NP-OSu) (Biosearch Technologies, Petaluma, CA). NP-chicken gamma globulin (NP-CGG) and NP-bovine serum albumin (NP-BSA) (Biosearch Technologies) were fluorescently labeled with Cyanine 5 (Cy5) on amine groups (Cy5 Mono-Reactive Dye, GE Healthcare).

C57BL/6 mice were purchased from Jackson Laboratories and housed in the Stanford Animal Facility for at least one week before use. *IL-17f*$^{Thy1.1/Thy1.1}$ mice and G8/*Rag2*$^{-/-}$ TCR transgenic mice were bred and housed in the pathogen-free Stanford Animal Facility. All experiments were performed in accordance with the Institutional Biosafety Committee and the Institutional Animal Care and Use Committee. 200 µg each of Cy3-CGG and CGG in aluminum hydroxide (Imject Alum; Thermo Scientific, Waltham, MA) per mouse and subcutaneous immunization were used in all studies.

### Antibodies and FACS analysis

Antibodies were purchased from either eBioscience or BD Biosciences unless otherwise stated. All analyses were performed on a BD LSR II flow cytometer. γδ T cells were enriched from mouse splenocytes by positive selection as described (*Jensen et al., 2008*). For NP experiments, staining of enriched γδ T cells was performed using 15 µg/ml NP$_{43}$-CGG-Cy5 or 0.02 µg/ml NP$_{67}$-PE or PE, along with PE or APC conjugated anti-TCRδ (GL-3), APC-Cy7 and Pacific Blue-labeled antibodies to αβ TCR (H57-597), B220 (RA2-6B2), F4/80 (BM8), Gr-1 (RB6-8C5), and CD11b (M1/70), and Aqua Amine live/dead stain (Invitrogen Molecular Probes, Eugene, OR). APC-Cy7, Pacific Blue, and Aqua positive cells were excluded from analysis. Anti-NP antibody (clone H33L γ; G. Kelsoe) Fab fragments were prepared using the Pierce Fab Preparation kit. For Cy3 experiments, enriched γδ T cells were stained with Cy3-conjugated protein (0.5 µM) on ice for 1 hr, along with APC conjugated GL-3, Aqua Amine, FITC conjugated antibodies to αβ TCR, B220, CD11b, CD11c (N418), Gr-1, and F4/80. FITC and Aqua-positive cells were excluded from the analysis.

For the analysis of CD44 expression, enriched γδ T cells were stained with FITC-conjugated antibody to CD44 (IM7), APC conjugated GL-3, and Cy3-OVA. For the analysis of Thy1.1 expression on cells isolated from *IL-17f*[Thy1.1/Thy1.1] reporter mice, enriched γδ T cells were stained with FITC conjugated antibody to Thy1.1 (OX-7; Biolegend), Pacific Blue conjugated antibody to CD62L (MEL-14), APC conjugated GL-3, and Cy3-OVA. Both analyses included the addition of Aqua Amine and APC-Cy7 labeled antibodies to αβ TCR, B220, CD11b, CD11c, Gr-1, and F4/80, with Aqua and APC-Cy7-positive cells excluded from analysis.

### Identification of antigen-specific γδ TCRs, in vitro stimulation assays and ligand binding to TCR expressed on cell surface

TCRs from Cy3- or NP-specific γδ T cells were identified at a single cell level and full length γ and δ TCR chain sequences were cloned and expressed in the 58α-β- cell line as described (*Shin et al., 2005*; *Zeng et al., 2012*). 58α-β- cells expressing γδ TCRs were stimulated with plate-bound NP- or Cy3-conjugates, the corresponding unmodified protein, or anti-CD3. The supernatant was collected and assayed for IL-2 production as described (*Zeng et al., 2012*).

Measurement of the kinetics of antigen binding to cell surface-expressed γδ TCR by real-time FACS analysis was carried out as described (*Zeng et al., 2012*). Briefly, 58α-β- cells expressing a Cy3-specific γδ TCR NX6 were stained with 40 nM $Cy3_4SAv$ for 1 hr at 4°C. Cells were spun down and resuspended in FACS buffer with 1600 nM anti-Cy3 (clone A-6, Santa Cruz Biotechnology, Dallas, TX) Fab, prepared using the Pierce Fab Preparation Kit. $Cy3_4SAv$ binding was recorded by flow cytometry over 1.5 to 3 hr at 5 or 10 min intervals (<10 s for each measurement). The data were fit using a first-order decay kinetic model to obtain the off-rate ($k_{off}$) and half-life ($t_{1/2}$).

Scatchard analysis of Cy3 binding to NX6/58α-β- cells was carried out as described in *Zeng et al. (2012)* with some modifications. $1 \times 10^5$ cells were incubated with 27.34–1.71 nM $Cy3_4SAv$. To quantify cell surface bound $Cy3_4SAv$, we biotinylated red blood cells (RBCs) to generate cells with different surface biotin densities (*Huang et al., 2010*). The same batch of biotinylated RBCs was stained with either PE-SAv or $Cy3_4SAv$. A linear correlation (*Figure 1—figure supplement 2*) between the mean fluorescence intensities of PE-SAv and $Cy3_4SAv$ was constructed, so that the $Cy3_4SAv$ staining intensities could be converted to PE-SAv intensities, which were used to calculate the number of bound ligands by comparing them with the standard PE calibration curve.

### Analysis of soluble TCR and ligand interactions

Soluble γδ TCRs were produced as described (*Zeng et al., 2012*). Briefly, the extracellular domains of the γ and δ chains (residues 1–273 and 1–242, respectively) were cloned in frame with a gene encoding a rhinovirus protease site, followed by acidic (TCR-δ) or basic (TCR-γ) leucine zippers and a (histidine)$_6$ tag in the pMSCV-P2 and Z4 retroviral expression vectors. These vectors contain an internal ribosome entry site followed by puromycin resistance gene for γ chain or zeocin resistance gene for δ chain and expressed in BHK-21 cells.

Surface plasmon resonance using the Biacore system was used for quantitative measurements of TCR-ligand interactions. All Biacore measurements were performed on a Biacore 3000 instrument using a CM5 chip. 10,000 RU of anti-TCRδ was immobilized using amine linkages; anti-TCRβ was immobilized as a reference surface. Roughly 300 RU of γδ TCR was injected into the system, allowed to stabilize for 1 min, then a range of concentrations of analytes were injected, followed by a 2 min dissociation time. For NP-specific TCRs, $NP_{43}$-CGG and CGG, were tested; for the Cy3-specific TCR, $Cy3_4$-SAv and streptavidin were tested. 10 mM glycine pH 2.5 was used at the end of each cycle to remove bound TCR and ligand. Specific binding was assessed by subtracting a blank buffer injection for each cycle. The dose response curves for $NP_{43}$-CGG, CGG, $Cy3_4$-SAv, and unconjugated SAv for specific binding were measured by averaging signal between 10 and 20 s at the end of each analyte injection, as very slow unbinding was observed.

### BioMark analysis

Quantitative analysis of transcript expression of Cy3-specific γδ T cells was carried out with the BioMark system as follows: 60 hr after Cy3-CGG immunization, γδ T cells were enriched from the draining lymph nodes of immunized mice, then incubated with Cy3-KLH (0.5 µM) for 6 hr in vitro. $CD62L^{lo}CD44^{hi}$ $Cy3^+$ and $CD62L^{hi}CD44^{lo}$ $Cy3^-$ γδ T cells were then FACS sorted into a PCR plate with five cells per well for the analysis. The primers for BioMark qPCR were purchased from Applied Biosystems. The sequences are described in *Supplementary file 1*.

Analyses of the expression data were performed with the R statistical package v.3.0.2. To compare the transcriptional profiles of Cy3$^+$ and Cy3$^-$ cells, we performed differential expression analysis using a two-sample Mann–Whitney test. Prior to hypothesis testing, we removed any gene that did not vary across the entire sample of Cy3$^+$ and Cy3$^-$ cells. Genes were considered significantly different at a Bonferroni-corrected p-value < 0.0019. Gene expression differences were displayed in a two-dimensional heatmap false colored based on transcript expression levels.

## Acknowledgements

We thank P Pereira for the Vγ1-specific antibody; G Kelsoe for the anti-NP monoclonal antibody; M Birnbaum for TCR purification advice.

## Additional information

### Funding

| Funder | Grant reference number | Author |
| --- | --- | --- |
| Office of Extramural Research, National Institutes of Health | R21AI107082 | Xun Zeng, Christina Meyer, Yueh-hsiu Chien |
| Office of Extramural Research, National Institutes of Health | RO1AI80829 | Xun Zeng, Christina Meyer, Jun Huang, Evan W Newell, Brian A Kidd, Yu-Ling Wei, Yueh-hsiu Chien |
| The Burt and Marion Avery Endowment | | Xun Zeng, Christina Meyer, Jun Huang, Evan W Newell, Brian A Kidd, Yu-Ling Wei, Yueh-hsiu Chien |
| Office of Extramural Research, National Institutes of Health | U19AI090019 | Yueh-hsiu Chien |

The funders had no role in study design, data collection and interpretation, or the decision to submit the work for publication.

### Author contributions

XZ, CM, JH, EWN, BAK, Y-LW, Conception and design, Acquisition of data, Analysis and interpretation of data, Revising the article; Y-C, Conception and design, Analysis and interpretation of data, Drafting and revising the article

### Author ORCIDs

Christina Meyer, http://orcid.org/0000-0002-0380-195X

### Ethics

Animal experimentation: All experiments involving animals were performed in accordance with the Institutional Biosafety Committee and the Institutional Animal Care and Use Committee (IACUC) of Stanford University. All animals were handled according to approved IACUC protocols (#9456 and #10081). Every effort was made to minimize suffering.

## Additional files

### Supplementary file

• Supplementary file 1. Primers used in this study.

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
