## [Decision Letter]

Thank you for sending your work entitled “γδ T cells recognize haptens and mount a hapten-specific response” for consideration at *eLife*. Your article has been favorably evaluated by Tadatsugu Taniguchi (Senior editor), a Reviewing editor, and 2 reviewers, one of whom, Craig Morita, has agreed to reveal his identity.

The Reviewing editor and the reviewers discussed their comments before we reached this decision, and the Reviewing editor has assembled the following comments to help you prepare a revised submission.

On the basis of the previous studies, the authors performed a series of experiments to convincingly demonstrate that murine γδ T cells can directly recognize two small organic molecules Cy3 and NP in a TCR-specific manner. Furthermore, activation of γδT cells was also investigated upon in vivo immunization with the haptens, and functionally, IL-17 was identified as the RoRgt-dependent cytokine produced by the hapten-specific γδT cells. The experiments are well done, and the data are very impressive and well presented, adding important new information about γδ T cell recognition of antigens. We and the reviewers have found this study is of interest, but further revision seems to be necessary, especially discussing high precursor frequency among γδ T cells for the various haptens in mice that have never been immunized with the hapten and what role of this biological characteristics of γδ T cells together with hapten-specific αβ T cells in contact sensitivity, as you can see from the comments attached. In addition, it's better to include a comparison with the recognition of small non-peptide molecules (pyrophosphjates) by human γδT cells.

Minor comments:

1) I suggest to broaden the Discussion a little bit and to include a comparison with the recognition of small non-peptide molecules (pyrophosphjates) by human gd T cells

2) The authors should show more or all of the BioMark data including the IFN-γ and IL-4 results in the figure as IFN-γ might be expected to be elevated in the non-Cy3 specific cells that are CD27 positive. There aren't that many probes used (46 genes).

---

## [Author Response]

We appreciate the opportunity to revise our manuscript. We have rewritten the Discussion section to include a comparison of our results with previous reports that human γδ T cells recognize small non-peptide molecules (pyrophosphates). We also discuss the antigen-specific γδ T cell and B cell frequencies in naïve animals. In fact, the frequency of hapten-specific γδ T cells is not higher than those of other antigen-specific γδ T cells. These γδ T cell antigens include nonclassical MHC class I molecules, T10 and T22, self antigens; and phycoerytherin (PE), an antigen from algae. The molecular basis for these relatively high frequencies of antigen-specific γδ T cells is not clear. Nonetheless, we demonstrated previously that the thymic maturation process does very little to constrain γδ T cell antigen specificity, but instead determines their effector fate (Jensen et al. ’08). Thus, the frequency of antigen-specific γδ T cells is largely determined by T cell receptor gene rearrangement. To further address the antigen-specific γδ T cell and B cell repertoires, we point out the work by Marc Jenkins and colleagues showing that in naïve mice, 0.1% of the B cells are PE-specific and 0.02% allophycocyanin (APC)-specific (Pape et al. Science ’11), and specify why earlier estimates of hapten-specific B cell frequencies may be too low.

We added new paragraphs discussing the possible biological significance of hapten-specific γδ T cells, including in the context of a hapten-driven pathological process, allergic contact dermatitis. We also revised Figure 2 to show all the negative results (i.e., no expression) from the BioMark experiments, including IFNγ and IL-4. Previously, we showed that PE immunization only activates PE-specific γδ T cells (Zeng et al. ’12); similarly, here we show that Cy3 immunization only activates Cy3-specific γδ T cells. Consistent with this observation that non-Cy3-specific γδ T cells are not activated, these cells do not make cytokines, IFNγ and IL-4 included.